# Test accuracy of polymerase chain reaction methods against conventional diagnostic techniques for Cutaneous Leishmaniasis (CL) in patients with clinical or epidemiological suspicion of CL: Systematic review and meta-analysis

**Luz Estella Mesa**[1,2], **Rubén Manrique**[2], **Carlos Muskus**[1], **Sara M. Robledo**[1]*

**1** PECET- Medical Research Institute, School of Medicine, University of Antioquia-UdeA, Calle, Medellín-Colombia, **2** Epidemiology and Biostatistic, Universidad CES Calle, Medellín-Colombia

* sara.robledo@udea.edu.co

## Abstract

### Background

Molecular diagnostic tests, notably polymerase chain reaction (PCR), are highly sensitive test for *Leishmania* detection, which is especially relevant in chronic cutaneous lesion with lower parasite load. An accurate diagnosis is essential because of the high toxicity of the medications for the disease. Nevertheless, diagnosis of cutaneous leishmaniasis (CL) is hampered by the absence of a reference standard. Assuming that the PCR-based molecular tools are the most accurate diagnostic method, the objective of this systematic review was to assess the diagnostic accuracy of PCR-based molecular tools in a meta-analysis of the published literature.

### Methodology/Principal findings

A search of the published literature found 142 papers of which only 13 studies met the selection criteria, including conventional PCR, real-time PCR, Loop-mediated isothermal amplification (LAMP), recombinase polymerase amplification (RPA), polymorphism-specific PCR (PS-PCR). The sensitivities of the individual studies ranged from 61% to 100%, and specificities ranged from 11% to 100%. The pooled sensitivities of PCR in smears were 0.95 (95% CI, 0.90 to 0.98), and the specificity was 0.91(95% CI, 0.70 to 0.98). In general population, estimates were lower in aspirates, skin biopsies and swab samples with 0.90 (95% CI, 0.80 to 0.95) and 0.87 (95% CI, 0.76 to 0.94) for sensitivity and specificity, respectively. The specificity was lower in consecutive studies, at 0.88 (95% CI, 0.59 to 0.98) and its CI were wider.

**Data Availability Statement:** All relevant data are within the manuscript and its Supporting Information files.

**Funding:** This work received financial support of the Colombian Administrative Department of Science, Technology and Innovation - Colciencias (Code: 617499847785; CT-695-2014). The funders had no role in study design, data collection and analysis, decision to publish, or preparation of the manuscript.

**Competing interests:** The authors have declared that no competing interests exist.

## Conclusions/Significance

No statistically significant differences between the accuracy in smears, aspirate, skin biopsies or swabs samples were observed. Therefore, a simple smear sample run by PCR, instead more invasive samples, may be enough to obtain a positive diagnosis of CL. The results for PCR in all samples type confirm previous reports that consider PCR as the most accurate method for the diagnosis of CL.

### Author summary

This systematic review and meta-analysis confirmed that PCR is the most accurate methods for the diagnosis of CL. The summary of the estimates for sensitivity and specificity in all readout methods of the index test were high. No statistically differences between the accuracy in smears, aspirate, skin biopsies or swabs samples suggesting that a simple smears sample run by PCR instead more invasive samples is enough to obtain a positive diagnosis of CL.

## Introduction

Leishmaniases are vector-borne infections caused by protozoa of genus *Leishmania*, affecting mammals. More than 30 *Leishmania* species are recognized, of which 20 are considered infective for humans and other mammals [1]. The ability to distinguish between *Leishmania* species is crucial for differentiation of the different clinical manifestations of the disease (visceral, cutaneous or mucocutaneus) to establish correct diagnosis and prognosis of the disease as well as to support decision-making regarding administration of the appropriate treatment. The rapid and accurate diagnosis of cutaneous leishmaniasis (CL) and the identification of the species involved in the infection are crucial for the therapeutic regimen and control of the disease.

The diagnosis of CL is based on the detection of the parasite in the sample collected directly from the patient´s lesions. The methods include direct microscope test, culture of aspirates and histopathology biopsies. Although the specificity of these methods is high, varying from 86% [1] to 100% [2], they have important drawbacks mainly related to their sensitivity and time consuming due to the high subjective component.

The direct test's sensitivity can vary depending of the type of tissue where the sample is collected, amount of sample obtained, as well as the technique used for the sampling and processing. The evolution time of the lesion and the previous use of treatments may also interfere with the sensitivity of the direct methods [3]. For all these reasons the sensitivity reported varies from 78.3% to 90.4% in samples taken from the active edge of the lesion *vs*. the base of the ulcer, respectively [4]. However, other studies have reported sensitivities as low as 32.7% and 37% in samples from the active edge of the lesion [5,6]. On the other hand, the culture-based and histopathology test has lower sensitivity than direct test and therefore they do not have an important role in the diagnosis of CL [3]. The inaccuracies in diagnosis prevent timely access to treatment and the establishment of guided strategies for the control and reduction of morbidity for leishmaniasis, affecting finally the patient welfare.

To overcome these drawbacks, different molecular tests are proposed for the diagnosis of CL by detecting the parasite genetic material (DNA or RNA), to improve (contrast) the accuracy of the traditional microscopic-based parasitological diagnosis. These molecular

techniques include multilocus enzyme electrophoresis (MLEE), conventional polymerase chain reaction (PCR) based assays, quantitative Real Time PCR or simplified PCR methods. In addition, available tools for species identification and phylogenetic analysis include DNA sequencing analysis, restriction fragment length polymorphism (RFLP) analysis, and PCR-fingerprinting techniques as well as novel methods such as multilocus sequence typing (MLST) and multi- locus microsatellite typing (MLMT).

The PCR based assays, are rapid, sensitive and discriminative at species or even strain level. They offer high flexibility and utility together with its sensitivity and specificity. Nevertheless, the complexity and cost are limiting factors for its routine application in clinics, restricting its use to research laboratory environments. However, there is an urgent need for standardization, optimization and simplification of PCR based applications to be used mainly in endemic areas around the world which will have an impact in disease control.

Providing clear evidence on the diagnostic accuracy of molecular tests allows to clarify the role of molecular techniques in epidemiological contexts. Here the importance of knowing the diagnostic accuracy of molecular tests so that public health institutions and those in charge of making decisions may implement the use of theses test for the control of CL. Based on the hypothesis that the PCR-based molecular tools are the most accurate diagnostic method, the objective of this systematic review was to assess the diagnostic attributes of PCR-based molecular tools in a meta-analysis of the published literature, with the purpose of contribute to the improvement of CL control.

## Methods

The review protocol was registered on PROSPERO 2017 and is available from: http://www.crd.york.ac.uk/PROSPERO/display_record.php?ID=CRD42017055859. This systematic literature review was performed based on recommendations by the Preferred Reporting Items for Systematic Reviews and Meta-Analyses (PRISMA) [7].

### Study scope and definition of the reference standard

This systematic review answers the question: what is the accuracy of the PCR-based methods in the diagnosis of CL in patients with potential infection attending health care services in endemic areas? For this purpose, the reference standard was defined as a compatible clinical lesion in addition to the demonstration of amastigotes through direct microscopic test or in combination with culture isolation of *Leishmania* parasites from lesion material. Other types of composite reference standards were not included. The confirmatory diagnostic of LC was based on the observation of intra- or extracellular *Leishmania* amastigotes through direct microscopic observation and/or promastigote isolation in cultures media.

### Literature search

To identify all relevant studies, we performed an electronic search using key terms in the following databases: PubMed, EMBASE and LILACS. These databases were selected because of their coverage of the literature and because we believe they will provide a representative sample of the published studies of diagnostic molecular tests from January 1990 to December 2018. In February 2019, The following MeSH terms were generated: "(((((((((((("Leishmaniasis, Cutaneous") OR "Leishmaniasis, Diffuse Cutaneous")) OR ("cutaneous leishmaniasis" OR "diffuse cutaneous leishmaniasis" OR "skin leishmaniasis"))) AND (("Polymerase Chain Reaction") OR ("Polymerase chain reaction" OR PCR))))) AND ((((((("Sensitivity and Specificity") OR "Sensitivity and Specificity/standards")) OR ("sensitivity OR specificity" OR "sensitivity OR specificity/standards" OR "sensitivity" OR "specificity"))) OR (screening OR "false positive" OR

"false negative" OR accuracy)) OR (((("Predictive Value of Tests") OR "Predictive Value of Tests/standards")) OR ("Predictive Value of Tests" OR "Predictive Value of Tests/standards" OR "predictive value" OR "predictive values of tests" OR "reference value" OR "references values"))) OR (("ROC Curve") OR ("ROC Curve" OR ROC "Receiver operating characteristics" OR "roc analysis" OR "roc analyses" OR "ROC and" OR "ROC area" OR "ROC auc" OR "ROC characteristics" OR "ROC curve method" OR "ROC curves" OR "ROC estimated" OR "ROC evaluation" OR "ROC likelihood ratio")))))) AND ("direct exam" OR "direct test" OR "direct tests" OR "direct microscopy" OR smear)) AND Humans). The limitations were the language (English/Spanish) and publication date (from 1990/01/01 to 2018/12/31), the search was limited to that period because it corresponds with the period where most of the molecular tests were developed. The search was also limited to those studies performed in humans.

## Inclusion criteria

In this review only prospective cohort studies were included (meaning, those studies including patients consecutively recruited, and those where all patients were submitted to the index test and the reference test), retrospective and cross-sectional studies; in some studies, re-counts of the same patients were performed to analyze different diagnostic techniques based on PCR (Table 1). All studies met the following criteria: (i) patients with clinical suspicion of cutaneous leishmaniasis, (ii) use of the molecular test for the diagnosis of CL, (iii) use of clinical samples isolated from humans, (iv) comparison with the reference standard "direct microscopic test alone or in combination with culture", (v) capacity of completion of a 2x2 contingency table. Those studies with patients suspicious of other leishmaniasis forms such as post kala-azar dermal leishmaniasis were excluded.

## Analysis of the selected papers

Two of the authors screened the titles and abstracts identified through the search strategy (SR-Professor of Immunology and LEM-PhD candidate student) and selected those studies potentially relevant for this review. For all relevant articles, the full text version was read to determine the presence of the inclusion criteria defined previously. Two independent reviewers read the full text articles (CM-Professor of Molecular Biology and LEM-PhD candidate student). In case of disagreement in any of the phases of evaluation, a third reviewer was consulted for the final decision (SR-Professor of Immunology).

A set of standard data from each study was collected using a data extraction form; two reviewers performed a pilot test with the initial form using 3 of the included publications; the data extraction form was modified and improved according to the information derived from the pilot test. In the case of studies where only one subgroup of participants was eligible, such as studies where different types of reference tests are analyzed simultaneously, only the data of the analysis comparing the reference standard defined previously were extracted.

Two reviewers independently extracted the data and completed the predefined data extraction form (CM and LEM), any disagreements were resolved through the discussion with a third reviewer (SR), and the extracted data included general information such as reference, study location, index tests, reference standard, sample, study type, control group, and any another relevant information.

We evaluated the methodological quality of the included studies using QUADAS-2 (Quality Assessment of Diagnostic Accuracy Studies). This tool is designed to assess the quality of primary diagnostic accuracy studies; consist of 4 key domains that discuss patient selection, index test, reference standard, and flow of patients through the study and timing of the index tests and reference standard (flow and timing); it is not designed to replace the data extraction

Table 1. Characteristics of the studies selected and test properties calculated after creating a 2 x 2 table.

| Reference | Yr | Index Tests | PCR Sample | Reference Standard[a] | TP | FP | FN | TN | Sensitivity (95% CI) | Specificity (95% CI) | PPV % (95% CI) | NPV % (95% CI) | Accuracy % (95% CI) | Study type | Control Group | Species/origin | Target | Primers | Amplicons size (bp) |
|---|---|---|---|---|---|---|---|---|---|---|---|---|---|---|---|---|---|---|---|
| Gunaratna et al. | 2018 | SpeedXtract-RPA | Punch biopsy, SSS, FNA | A | 61 | 0 | 32 | 57 | 65.59 (55.02–75.14) | 100 (93.74–100) | 100 (94.13–100) | 64.04 (53.18–73.95) | 78.67 (71.24–84.93) | Consecutive | NA | L. donovani | kDNA | FP3/RP3 | 160 |
| León CM et al. | 2018 | LAMP | Smears | A | 36 | 4 | 0 | 10 | 100 (90.26–100) | 71.43 (41.90–91.61) | 90 (76.34–97.21) | 100 (69.15–100) | 92 (80.77–97.78) | Consecutive | NA | Unknown | 18S rRNA | F3;B3;FIP; BIP | <120[c] |
| Koltas et al (a) | 2016 | kDNA-PCR | Smears | A | 30 | 11 | 0 | 31 | 100 (88.43–100) | 73.81 (57.96–86.14) | 73.17 (57.06–85.78) | 100 (88.78–100) | 84.72 (74.31–92.12) | Consecutive | NA | L. tropica; L. infantum; L. major | kDNA | 13A/13B | 120 |
| Koltas et al (b) | 2016 | SSU rRNA-PCR | Smears | A | 30 | 8 | 0 | 34 | 100 (88.43–100) | 80.95 (65.88–91.40) | 78.95 (62.68–90.44) | 100 (89.72–100) | 88.89 (79.28–95.08) | Consecutive | NA | L. tropica; L. infantum; L. major | SSU rRNA | R221/R332 | 603 |
| Koltas et al (c) | 2016 | ITS2-PCR | Smears | A | 30 | 4 | 0 | 38 | 100 (88.43–100) | 90.48 (77.38–97.34) | 88.24 (72.55–96.70) | 100 (90.75–100) | 94.44 (86.38–98.47) | Consecutive | NA | L. tropica; L. infantum; L. major | ITS2 | L5.8SR/LISTV | 700 |
| Koltas et al (d) | 2016 | ITS1-PCR | Smears | A | 29 | 0 | 1 | 42 | 96.67 (82.78–99.92) | 100 (91.59–100) | 100 (88.06–100) | 97.67 (87.71–99.94) | 98.61 (92.50–99.97) | Consecutive | NA | L. tropica; L. infantum; L. major | ITS1 | L58S/LITSR | 320 |
| Koltas et al (e) | 2016 | ME-PCR | Smears | A | 27 | 0 | 3 | 42 | 90 (73.47–97.89) | 100 (91.59–100) | 100 (87.23–100) | 93.33 (81.73–98.60) | 95.83 (88.30–99.13) | Consecutive | NA | L. tropica; L. infantum; L. major | ME | FME/RME | 450 |
| Koltas et al (f) | 2016 | HSP70-PCR | Smears | A | 26 | 0 | 4 | 42 | 86.67 (69.28–96.24) | 100 (91.59–100) | 100 (86.77–100) | 91.30 (79.21–97.58) | 94.44 (86.38–98.47) | Consecutive | NA | L. tropica; L. infantum; L. major | HSP70 | HSP70sen/HSP70ant | 1422 |
| Abd El-Salam et al. | 2014 | kDNA PCR | Smears | A | 53 | 46 | 8 | 6 | 86.89 (75.78–94.16) | 11.54 (4.35–23.44) | 53.54 (43.23–63.62) | 42.86 (17.66–71.14) | 52.21 (42.61–61.70) | Consecutive | NA | L. tropica | kDNA | kDNA-Fw/kDNA-Rv[b] | 186 |
| Eroglu et al. (a) | 2014 | Real-time PCR | Smears | A | 40 | 26 | 2 | 36 | 95.24 (83.84–99.42) | 58.06 (44.85–70.49) | 60.60 (47.81–72.42) | 94.74 (82.25–99.36) | 73.08 (63.49–81.31) | Consecutive | NA | L. tropica; L. infantum; L. major | ITS1 | LITSR/L.5.8S | 300–350 |
| Eroglu et al. (b) | 2014 | PCR | Smears | A | 39 | 24 | 3 | 38 | 92.86 (80.52–98.50) | 61.29 (48.07–73.40) | 61.90 (48.80–73.85) | 92.68 (80.08–98.46) | 74.04 (64.52–82.14) | Consecutive | NA | L. tropica; L. infantum; L. major | kDNA | 13A/13B | 120 |
| Adams et al. (a) | 2014 | Qiagen (qPCR) | Swab | B | 78 | 4 | 2 | 21 | 97.50 (91.26–99.70) | 84 (63.92–95.46) | 95.12 (87.98–98.66) | 91.30 (71.96–98.93) | 94.29 (87.98–97.87) | Consecutive | NA | L. panamensis | 18S rDNA | rDNA-Fw/rDNA-Rv[b] | 61 |
| Adams et al. (b) | 2014 | Isohelix (qPCR) | Swab | B | 74 | 4 | 6 | 21 | 92.50 (84.39–97.20) | 84 (63.92–95.46) | 94.87 (87.39–98.59) | 77.78 (57.74–91.38) | 90.48 (83.18–95.34) | Consecutive | NA | L. panamensis | 18S rDNA | rDNA-Fw/rDNA-Rv[b] | 61 |
| Adams et al. (c) | 2014 | Qiagen (qPCR) | Aspirate | B | 64 | 2 | 16 | 23 | 80 (69.56–88.11) | 92 (73.97–99.02) | 96.97 (89.48–99.63) | 58.97 (42.10–74.43) | 82.86 (74.27–89.51) | Consecutive | NA | L. panamensis | 18S rDNA | rDNA-Fw/rDNA-Rv[b] | 61 |
| Adams et al. (d) | 2014 | Boil/Spin (qPCR) | Aspirate | B | 49 | 1 | 31 | 24 | 61.25 (49.70–71.94) | 96 (79.65–99.90) | 98 (89.35–99.95) | 43.64 (30.30–57.68) | 69.52 (59.78–78.13) | Consecutive | NA | L. panamensis | 18S rDNA | rDNA-Fw/rDNA-Rv[b] | 61 |
| El-Beshbishy et al. (a) | 2013 | kDNA PCR (seminested) | Skin biopsies | B | 24 | 3 | 2 | 5 | 92.31 (74.87–99.05) | 62.50 (24.49–91.48) | 88.89 (70.84–97.65) | 71.43 (29.04–96.33) | 85.29 (68.94–95.05) | Consecutive | NA | L. major; L. tropica | kDNA | kDNA-Fw/kDNA-Rv[b] | 757 |

(Continued)

**Table 1.** (Continued)

| Reference | Yr | Index Tests | PCR Sample | Reference Standard[a] | TP | FP | FN | TN | Sensitivity (95% CI) | Specificity (95% CI) | PPV % (95% CI) | NPV % (95% CI) | Accuracy % (95% CI) | Study type | Control Group | Species/origin | Target | Primers | Amplicons size (bp) |
|---|---|---|---|---|---|---|---|---|---|---|---|---|---|---|---|---|---|---|---|
| El-Beshbishy et al. (b) | 2013 | ITS1-PCR | Skin biopsies | B | 18 | 0 | 8 | 8 | 69.23 (48.21–85.67) | 100 (63.06–100) | 100 (81.47–100) | 50 (24.65–75.35) | 76.47 (58.83–89.25) | Consecutive | NA | L. major; L. tropica | ITS1 | LITSR/L5.8S | 321 |
| Marco et al. | 2012 | PS-PCR | Smears | A | 36 | 3 | 8 | 15 | 81.82 (67.29–91.80) | 83.33 (58.58–96.42) | 92.31 (79.13–98.38) | 65.22 (42.73–83.62) | 82.26 (70.47–90.80) | Case-control | non-ATL cases | L. braziliensis; L. guyanensis; L. panamensis | PS | V1/V2; M1/M2 | 168; 700 |
| Khosravi et al. | 2012 | Real-time PCR | Skin biopsies | B | 67 | 13 | 1 | 19 | 98.53 (92.07–99.96) | 59.38 (40.64–76.30) | 83.75 (73.82–91.05) | 95 (75.13–99.87) | 86 (77.63–92.13) | Consecutive | NA | L. major; L. tropica | TRYP | TRYP-F/TRYP-R | 91 |
| Meymandi et al. | 2009 | Real-time PCR | Smears | A | 30 | 2 | 0 | 14 | 100 (88.43–100) | 87.50 (61.65–98.45) | 93.75 (79.19–99.23) | 100 (76.84–100) | 95.65 (85.16–99.47) | Case-control | Individuals clinically suspected but without Leishman bodies | Unknown | kDNA | 13A/13B | 120 |
| Lemrani et al. | 2009 | SSU rRNA-PCR | Skin biopsies | B | 11 | 0 | 2 | 5 | 84.62 (54.55–98.08) | 100 (47.82–100) | 100 (71.51–100) | 71.43 (29.04–96.33) | 88.89 (65.29–98.62) | Case-control | Patients with other skin diseases similar to CL | L. infantum; L. tropica; L. major | SSU rRNA | R221/R332 | 650 |
| Kumar et al. (a) | 2007 | ITS1-PCR | Skin biopsies | B | 24 | 0 | 5 | 3 | 82.76 (64.23–94.15) | 100 (29.24–100) | 100 (85.75–100) | 37.5 (8.52–75.51) | 84.38 (67.21–94.72) | Consecutive | NA | L.tropica | ITS1 | LITSR/L5.8S | 300–350 |
| Kumar et al. (b) | 2007 | kDNA PCR | Skin biopsies | B | 28 | 1 | 1 | 2 | 96.55 (82.24–99.91) | 66.67 (9.43–99.16) | 96.55 (82.24–99.91) | 66.67 (9.43–99.16) | 93.75 (79.19–99.23) | Consecutive | NA | L.tropica | kDNA | Uni21/Lmj4 | 850; 650 |
| Al-Jawabreh et al. | 2006 | ITS1-PCR | Smears | A | 52 | 0 | 8 | 45 | 86.67 (75.41–94.06) | 100 (92.13–100) | 100 (93.15–100) | 84.91 (72.41–93.25) | 92.38 (85.54–96.65) | Case-control | Healthy volunteers | L.major; L. tropica | ITS1 | LITSR/L5.8S | 319–335 |

TP: true positives; FP: false positives; FN: false negatives; TN: true negatives; CI: confidence interval; PPV: positive predictive value; NPV: negative predictive value. RPA: recombinase polymerase amplification; LAMP: Loop-mediated isothermal amplification; kDNA: kinetoplast DNA; SSU rRNA: small subunit rRNA; ITS: internal transcribed spacer; ME: mini exon; HSP70: heat-shock protein; PCR: polymerase chain reaction; qPCR: quantitative PCR; PS-PCR: polymorphism-specific PCR; SSS: slit skin smear; FNA: fine needle aspirate; NA: not applicable; TRYP: Tryparedoxim peroxidase gene.

Publication with more than one amplification method were subdivided.

[a] Standard A, Direct Microscopic; Standard B, Direct microscopic and/or Culture.

[b] The author did not identify the primers in the article.

[c] Based on the place where the primers hybridize, but amplicons vary depending on the number of loops that are formed.

process but complement the data extraction process of a systematic review. The researchers involved in the data extraction process were trained in the use of the QUADAS-2 tool [8].

Data extraction and quality assessment were done independently by two reviewers (CM and LEM). Any discrepancies were resolved by consulting (SR). For 10% of the included studies, data extraction was also done by (SR). The decision regarding which studies would be included in the meta-analysis was primarily resolved using criteria related to the test methodology.

### Anticipated sources of heterogeneity

First, the selected papers were separated according to the reference standard used (direct microscopic test alone or in combination with culture) to define diseased and non-diseased subjects. Secondly, the type of PCR sample analyzed, study design, polymerase chain reaction target, size of the amplicons (pb), primes were considered. The DNA extraction methods that used commercial kits or phenol-extraction techniques were considered similar for human tissue samples that were stored in filter paper, paraffin embedded or frozen [9]; in addition, the studies that used primers that amplify multiple copy genes were considered similar too [9].

Searching to avoid heterogeneity among diagnostic test accuracy (DTA) studies, was focused on selecting a more homogeneous set of studies, however, heterogeneity is great among DTA studies and it is often difficult to strictly control biases. In order to achieve accurate review results, we used a first subgroup analysis that included studies that used exclusively the direct microscopic test as a reference standard. A second subgroup analysis was done with those studies where the reference standard included both direct microscopic test and/or culture but with comparable or similar methodologies.

### Statistical analysis

For all studies, estimates of sensitivity, specificity, positive and negative predictive value and 95% confidence intervals (CI) were expressed in forest plots in Review Manager version 5.3. We therefore investigated sources of heterogeneity by adding the following covariates to the mixed-effects logistic regression model: (i) the index tests, (ii) PCR sample, (iii) the study design, (iv) the reference standard and (v) the target gene (single or multicopy gene) using the command *xtmelogit* in STATA version 14. A covariate was assumed to have a significant effect on the estimates of sensitivity and specificity and thus to explain some of the heterogeneity in the studies included in meta-analysis if the $P$ value was $< 0.05$. We used two methods of meta-analyzing diagnostic accuracy data, which are statistically rigorous: the hierarchical summary receiver operating characteristic (HSROC) model [10] and the bivariate logit-normal random-effects meta-analysis model [11] to obtain a summary estimate of sensitivity and specificity, statistical analysis using *xtmelogit* in STATA version 14. Diagnostic accuracies between various readout methods for PCR (index tests), PCR samples, the study design and reference standard, were compared using Chi-square test with the programme STATA version 14.

## Results

### Included studies

The electronic search yielded 142 results, 42 of which were taken forward to read the full text (Fig 1). Articles were excluded at this stage due to (i) articles not available, (ii) not in the field of interest, (iii) use of an inappropriate reference standard, and (iv) inability to complete a 2-by-2 contingency table. A total of 13 articles were included in the systematic review [5,12–23] while data of 12 articles were included in the meta-analysis. Data of 1 article were not

## PRISMA 2009 Flow Diagram

**Identification**

142 Records identified through database searching

No additional records identified through other sources

**Screening**

92 Records after duplicates removed

92 Records screened

50 Records excluded after reading title/abstract

**Eligibility**

42 Full-text articles assessed for eligibility

29 Full-text articles excluded, with reasons:
Not in the field of interest (5)
Articles not available (2)
Inappropriate reference standard* (15)
Cannot make 2-by-2 table (7)

**Included**

13 Studies included in qualitative synthesis

12 Studies included in quantitative synthesis (meta-analysis)

\* That no adjusted to the definition of the reference standard previously established.

**Fig 1. Flow chart of included studies.**

considered in the meta-analysis because they did not fit the respective subgroups for molecular method and/or sample type. Data extracted by a third reviewer as a quality assurance procedure was in agreements with the data extracted by the primary reviewers.

In the included studies, the index tests assessed were RPA ($n = 1$), LAMP ($n = 1$), conventional PCR ($n = 14$), real-time PCR ($n = 3$), qPCR ($n = 4$), PS-PCR ($n = 1$). Studies performing DNA extraction from smears ($n = 13$), SSS (slit skin smear) ($n = 1$), FNA (fine needle aspirate) ($n = 1$), punch biopsy ($n = 1$), swab ($n = 2$), aspirate ($n = 2$), skin biopsies ($n = 6$).

## Quality assessment of study reports

Although we may choose to restrict the primary analysis to include only studies at low risk of bias or with low concern about applicability for either all or specified domains, we considered that is often preferable to review all relevant evidence and then investigate possible sources of heterogeneity. The results of quality assessment with the QUADAS-2 are summarized in terms of risk of bias and concerns regarding applicability for all included studies (Fig 2). Of the 13 studies included, 4 were case-control studies, which resulted in a considerable proportion of studies having high risk of bias and high concern regarding applicability in the domain of patient selection. Respect to if could the conduct or interpretation of the index test have introduced bias, did raise problems in a large proportion of studies 85% (11/13), however there were no concerns regarding applicability in all the studies for this domain. A reference standard with unclear risk of bias was used in approximately 85% of the studies (11/13); even so, the target condition, as defined by the reference standard, was applicable to our review in all the studies. In 10 out of 13 studies, the patient's flow could have introduced bias; there was no details about that the results of the index test and reference standard were collected on the same patients at the same time.

## Diagnostic accuracy of molecular tests and analysis of heterogeneity

The 13 articles included 24 separate studies, that mean that more than one index test evaluated per article, for these 24 studies 2-by-2 contingency tables could be completed. Sensitivities ranged from 61% to 100%, and specificities ranged from 11% to 100% (Table 1). A coupled forest plot of sensitivity and specificity of PCR are shown in Fig 3. Since it is important to take into account the type of sample used to diagnose CL, we decided that this analysis would be about the type of sampling procedure used; in this order, the summary HSROC curve for PCR in smear sample, reference standard A and case control and consecutive studies are show in Fig 4A, and the

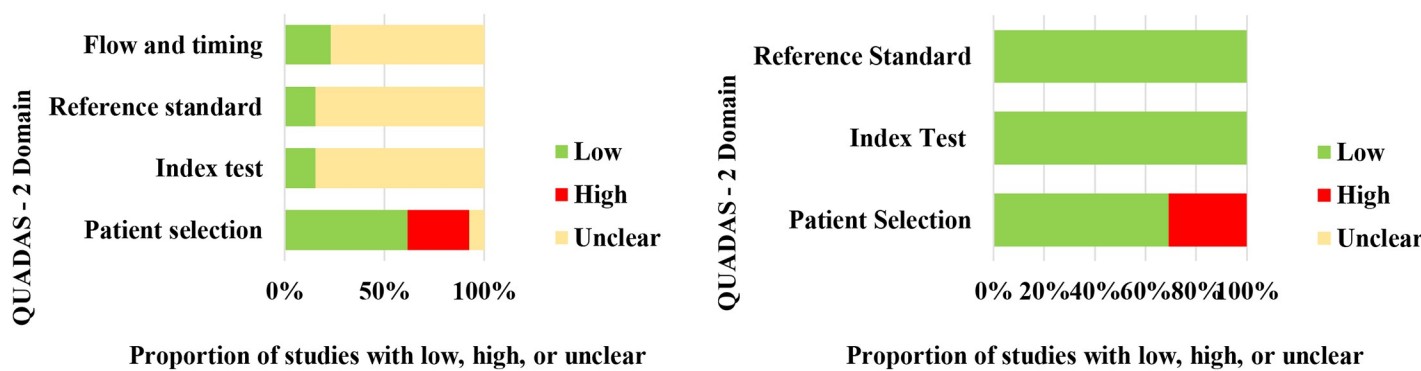

**Fig 2. QUADAS-2 results showing risk of bias and applicability concerns of the studies selected.**

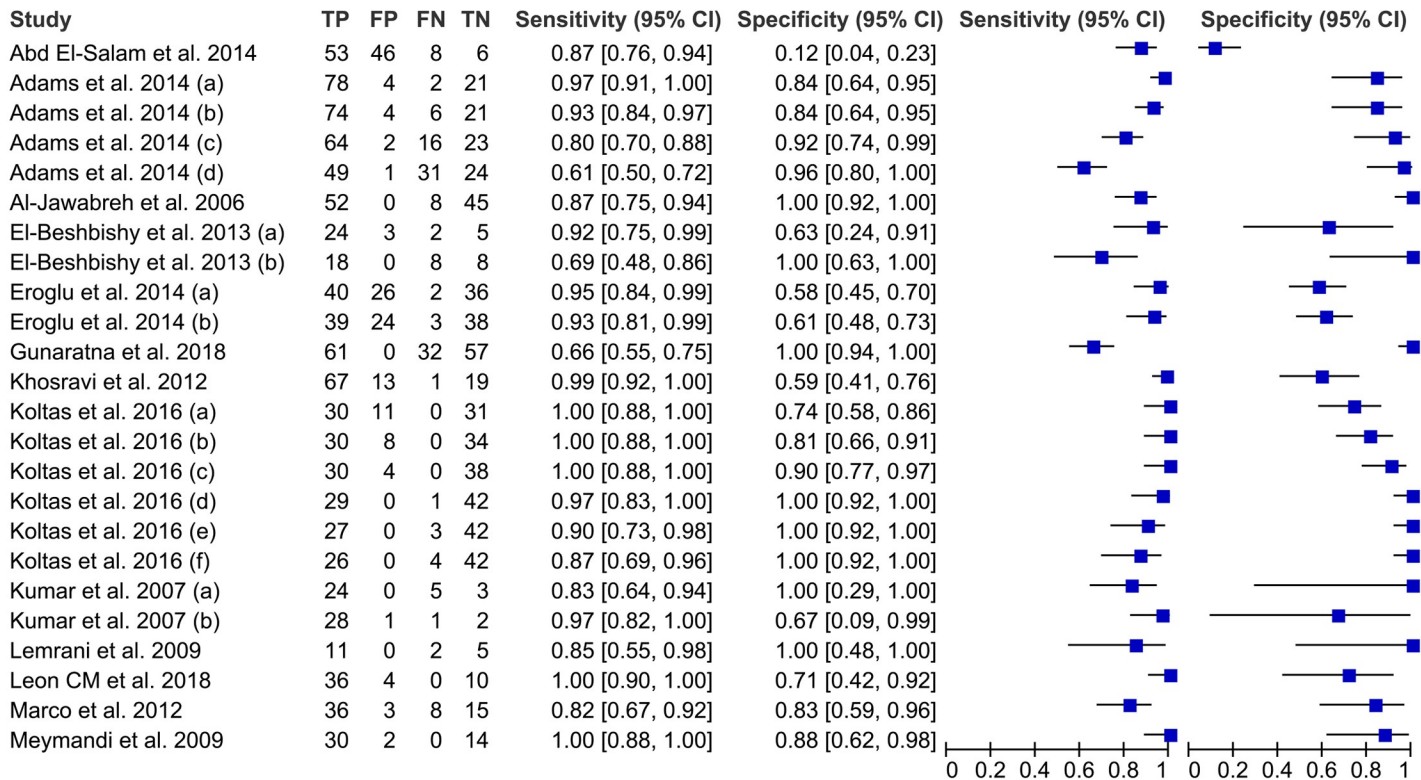

| Study | TP | FP | FN | TN | Sensitivity (95% CI) | Specificity (95% CI) |
|---|---|---|---|---|---|---|
| Abd El-Salam et al. 2014 | 53 | 46 | 8 | 6 | 0.87 [0.76, 0.94] | 0.12 [0.04, 0.23] |
| Adams et al. 2014 (a) | 78 | 4 | 2 | 21 | 0.97 [0.91, 1.00] | 0.84 [0.64, 0.95] |
| Adams et al. 2014 (b) | 74 | 4 | 6 | 21 | 0.93 [0.84, 0.97] | 0.84 [0.64, 0.95] |
| Adams et al. 2014 (c) | 64 | 2 | 16 | 23 | 0.80 [0.70, 0.88] | 0.92 [0.74, 0.99] |
| Adams et al. 2014 (d) | 49 | 1 | 31 | 24 | 0.61 [0.50, 0.72] | 0.96 [0.80, 1.00] |
| Al-Jawabreh et al. 2006 | 52 | 0 | 8 | 45 | 0.87 [0.75, 0.94] | 1.00 [0.92, 1.00] |
| El-Beshbishy et al. 2013 (a) | 24 | 3 | 2 | 5 | 0.92 [0.75, 0.99] | 0.63 [0.24, 0.91] |
| El-Beshbishy et al. 2013 (b) | 18 | 0 | 8 | 8 | 0.69 [0.48, 0.86] | 1.00 [0.63, 1.00] |
| Eroglu et al. 2014 (a) | 40 | 26 | 2 | 36 | 0.95 [0.84, 0.99] | 0.58 [0.45, 0.70] |
| Eroglu et al. 2014 (b) | 39 | 24 | 3 | 38 | 0.93 [0.81, 0.99] | 0.61 [0.48, 0.73] |
| Gunaratna et al. 2018 | 61 | 0 | 32 | 57 | 0.66 [0.55, 0.75] | 1.00 [0.94, 1.00] |
| Khosravi et al. 2012 | 67 | 13 | 1 | 19 | 0.99 [0.92, 1.00] | 0.59 [0.41, 0.76] |
| Koltas et al. 2016 (a) | 30 | 11 | 0 | 31 | 1.00 [0.88, 1.00] | 0.74 [0.58, 0.86] |
| Koltas et al. 2016 (b) | 30 | 8 | 0 | 34 | 1.00 [0.88, 1.00] | 0.81 [0.66, 0.91] |
| Koltas et al. 2016 (c) | 30 | 4 | 0 | 38 | 1.00 [0.88, 1.00] | 0.90 [0.77, 0.97] |
| Koltas et al. 2016 (d) | 29 | 0 | 1 | 42 | 0.97 [0.83, 1.00] | 1.00 [0.92, 1.00] |
| Koltas et al. 2016 (e) | 27 | 0 | 3 | 42 | 0.90 [0.73, 0.98] | 1.00 [0.92, 1.00] |
| Koltas et al. 2016 (f) | 26 | 0 | 4 | 42 | 0.87 [0.69, 0.96] | 1.00 [0.92, 1.00] |
| Kumar et al. 2007 (a) | 24 | 0 | 5 | 3 | 0.83 [0.64, 0.94] | 1.00 [0.29, 1.00] |
| Kumar et al. 2007 (b) | 28 | 1 | 1 | 2 | 0.97 [0.82, 1.00] | 0.67 [0.09, 0.99] |
| Lemrani et al. 2009 | 11 | 0 | 2 | 5 | 0.85 [0.55, 0.98] | 1.00 [0.48, 1.00] |
| Leon CM et al. 2018 | 36 | 4 | 0 | 10 | 1.00 [0.90, 1.00] | 0.71 [0.42, 0.92] |
| Marco et al. 2012 | 36 | 3 | 8 | 15 | 0.82 [0.67, 0.92] | 0.83 [0.59, 0.96] |
| Meymandi et al. 2009 | 30 | 2 | 0 | 14 | 1.00 [0.88, 1.00] | 0.88 [0.62, 0.98] |

**Fig 3. Couplet forest plots of the studies selected in Review Manager 5.3.**

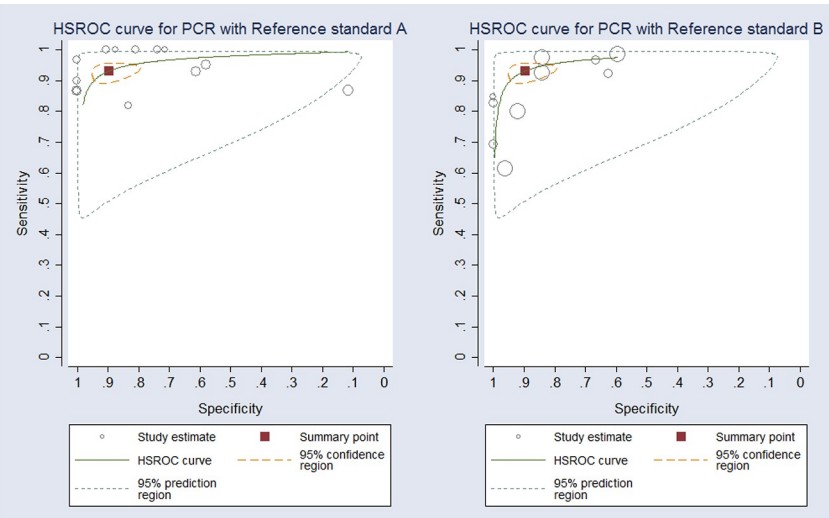

**Fig 4. HSROC curve for PCR. A.** In smear sample, reference standard A, case control and consecutive studies combined. Circles represent estimates of individual primary studies, and square indicates summary points of sensitivity and specificity. The circled region around the solid square represent the 95% CI region around the summary estimate. HSROC curve is obtained using command "metandiplot" in STATA version 14. **B.** In skin biopsies and aspirate samples, reference standard B and case control and consecutive studies combined. Circles represent estimates of individual primary studies, and square indicates summary points of sensitivity and specificity. The circled region around the solid square represent the 95% CI region around the summary estimate. HSROC curve is obtained using command "metandiplot" in STATA version 14.

HSROC curve for PCR in aspirate, skin biopsies and swab, reference standard B and case control and consecutive studies are show in Fig 4B. There was no statistically significant difference in accuracy (sensitivity and specificity) between the various readout methods for PCR (LAMP, conventional PCR, real-time PCR, qPCR, PS-PCR), allowing the results to be pooled in the analysis (P value 0.469; 95% CI: -.613–1.331). There was no also statistically significant differences in accuracy between smears, SSS, FNA, punch biopsy, swab, aspirate and skin biopsies samples (P value 0.058; 95% CI: -.015-.940). Similarly, there was no statistically significant difference in accuracy between consecutive and case-control studies (P value 0.610; 95% CI: -.707–1.205). There was no statistically significant differences in accuracy between reference standard A and B (P value 0.537; 95% CI: -1.047–0.545) and there was no statistically significant differences in accuracy between multi and single copy target genes (P value 0.884; 95% CI: -0.974–0.839). When the results of the case-control studies were compared with those of the consecutive studies, which used as a reference standard direct microscopic test, no difference for sensitivity and specificity was found. The sensitivity and specificity in the studies which used as a reference standard direct microscopic and/or culture was slightly lower than in the studies that used reference standard test alone (Table 2).

The summary estimates for sensitivity and specificity for both "All readout methods of the index test except LAMP and Real-time PCR" and "All readout methods of the index test without exception" on smears samples (general population) were high; however, the estimates were lower in aspirate, skin biopsies and swab samples in general population. The consecutive studies still show high summary sensitivities of 0.95 for smears (All readout methods of the index test except LAMP and Real-time PCR) and 0.96 (All readout methods of the index test without exception), but specificities are lower at 0.93 and 0.88, respectively and their CI are wider.

**Table 2. Summary estimates for PCR[a].**

| Test and sample type | Reference standard[b] | No. of articles (No. studies)[c] | Pooled sensitivity (95% CI)[d] | Pooled specificity (95% CI)[d] |
|---|---|---|---|---|
| Case-control and consecutive studies combined | | | | |
| PCR smears[e] | A | 5(10) | 0.93 (0.87–0.96) | 0.95 (0.68–0.99) |
| PCR smears[f] | A | 7(13) | 0.95 (0.90–0.98) | 0.91 (0.70–0.98) |
| Consecutive studies | | | | |
| PCR smears[e] | A | 3(8) | 0.95 (0.88–0.98) | 0.93 (0.56–0.99) |
| PCR smears[f] | A | 4(10) | 0.96 (0.90–0.98) | 0.88 (0.59–0.98) |
| Case-control and consecutive studies combined | | | | |
| PCR aspirate, skin biopsies and swab[e] | B | 4(9) | 0.88 (0.77–0.93) | 0.90 (0.80–0.95) |
| PCR aspirate, skin biopsies and swab[f] | B | 5(10) | 0.90 (0.80–0.95) | 0.87 (0.76–0.94) |

[a] Statistical analysis in STATA version 14, is obtained using command "xtmelogit".

[b] Standard A, Direct Microscopic; Standard B, Direct microscopic and/or Culture.

[c] The 12 articles included 23 separate studies, that means that more one index test evaluated per article.

[d] CI, confidence interval.

[e] All readout methods of the index test except LAMP and Real-time PCR

[f] All readout methods of the index test without exception

Further analysis was confined to the smears samples in the studies, which used as a reference standard the direct microscopic test, given, that the subgroup that used aspired, skin biopsies and swab samples in studies with reference standard direct microscopic and/or culture were too small to perform an analysis of heterogeneity. Furthermore, there was no big differences in accuracy between studies that used direct microscopy and, studies that used direct microscopy and/or culture as a reference standard.

Since there is no recommended method to assess the publication bias and that no inferences can be made regarding the presence or absence of this bias, in this systematic review and meta-analysis was not assessed the publication bias. Nevertheless, we recognize that this bias is a serious problem which can affect the validity and generalization of conclusions.

## Discussion

Molecular test with high quality sensitivity has been widely used for diagnosing of CL. PCR is suitable when there are atypical lesion of CL and few numbers of parasites, or when the microscopic method is negative, molecular diagnosis appears to be the solution to the short-falls on traditional diagnostic methods. In this systematic review, we analyzed and summarized data from diagnostic accuracy studies of molecular test in the diagnosis of CL. After to identify all relevant studies from the available literature, we were able to assess the accuracy of PCR tests in smears, aspirate, skin biopsies and swab.

The finding the high estimates for sensitivity and specificity on smears samples and low estimates on aspirate, skin biopsies and swab samples in general population, allows us to believe that a simple smears sample would suffice instead of taking more invasive skin biopsies or aspirate samples. This finding is important from patient's point of view because highlighting the use of non-invasive sampling procedures, which generate greater stigmatization in patients due to the permanent scars.

The low specificities found in all readout methods of the index test, can be attributed, in part, to the fact that the controls in case-control studies are often healthy persons, whereas controls in consecutive studies are in fact suspected patients; variation in the controls were one of the most frequent points of heterogeneity among the studies [8]. The high number of positive PCR tests in suspected patients with a negative reference standard may be explained by a proportion of the false positives being true positive when we take into consideration that the reference standard for CL is imperfect and that the sensitivity of PCR test is superior to the reference standard.

Consecutive studies better reflect the diagnosis situation and are thus of higher methodological quality than case-control studies [23]. We therefore recommend that future diagnostic accuracy studies should use a consecutive design to determine whether our findings about specificity are reproducible and to obtain valid estimates.

This meta-analysis shows that the molecular methods are very sensitive tools for the detection of *Leishmania* parasites in smears samples instead of invasive skin biopsies or aspirate samples, for which, we found lower pooled estimated for sensitivity, as well as specificity. Though due to limited data, it was not possible to provide summary estimates for consecutive studies and case-control studies separately on aspirate, skin biopsies and swab samples. Regarding to the future diagnostic accuracy studies, we highlight the fact of leave out the case-control studies to avoid that diagnostic accuracy being overestimated and consider the results for PCR in simple smears sample such as the most accurate method for the diagnosis of CL instead more invasive samples [24].

The results of this systematic review and meta-analysis have important implications in public health because the diagnostic of CL through direct microscopic test is a highly operator

dependent method. This means that the results depend of skills of who performs the test; therefore, the molecular methods can help to avoid this bias and permit obtain improve diagnostics and ensure the adequate treatments in patients. It is also important that the sensitivity of the molecular methods for the diagnosis of CL be high in a simple smear samples instead of invasive samples, because the scars left by the disease already stigmatize the patients and obtaining a sample through invasive methods would be useless.

## Conclusions

Results suggest that the molecular methods are very sensitive tools for the detection of *Leishmania* parasites in smears samples instead of invasive skin biopsies or aspirate samples, we no found statistically differences between the accuracy in smears, aspirate, skin biopsies or swabs samples. Therefore, we consider that a simple smear sample run by PCR instead more invasive sample is enough to obtain a positive diagnosis of CL. The results for PCR in all samples type confirm previous reports that consider PCR as the most accurate method for the diagnosis of CL.

## Limitations

Many studies in our meta-analysis suffer from poor quality. Thirty-one percent of the included studies had a case-control study type and this design is reputed to introduce selection bias, as the cases are confirmed patients and the controls are healthy volunteers or patients with other skin diseases similar to CL. Imperfect reference standard bias is an important issue in diagnostic accuracy studies. In the case of CL, there is a risk of underestimates the specificity of a new test when comparing it to current methods that have low sensitivity and high specificity, such as reference standard.

QUADAS-2 assessment showed that 85% of the studies, the conduct or interpretation of the index test and the reference standard could have introduced bias when judged against these definitions.

The analysis of heterogeneity done or all studies combined did not show a significant difference between studies that used standard A, reference standard and studies that used a composite reference standard B, reference standard and/or culture. Neither were significant differences between the readout methods of the index test. Nevertheless, subgroups were analyzed separately: "all readout methods of the index test except LAMP and real-time PCR" and "all readout methods of the index test without exception".

The lack of standardization is another limitation when you want to compare diagnostic accuracy studies for molecular tools. These different protocols are evidenced in the Table 1. An additional problem encountered during all process of studies selection, data extraction, and quality assessment of the included studies, were incomplete reporting of studies. This is due to the not use of the STARD guidelines for reporting diagnostic accuracy studies.

## Supporting information

**S1 Checklist. PRISMA-DTA for abstract checklist.**
(DOCX)

**S2 Checklist. PRISMA-DTA checklist.**
(DOCX)

## Author Contributions

**Conceptualization:** Luz Estella Mesa, Sara M. Robledo.

**Data curation:** Luz Estella Mesa.

**Formal analysis:** Luz Estella Mesa, Rubén Manrique.

**Funding acquisition:** Sara M. Robledo.

**Methodology:** Luz Estella Mesa.

**Supervision:** Sara M. Robledo.

**Validation:** Luz Estella Mesa.

**Writing – original draft:** Luz Estella Mesa.

**Writing – review & editing:** Luz Estella Mesa, Rubén Manrique, Carlos Muskus, Sara M. Robledo.

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
