## [Decision Letter · Decision Letter 0]

21 Oct 2019

Dear Professor Robledo:

Thank you very much for submitting your manuscript "Test accuracy of polymerase chain reaction methods against conventional diagnostic techniques for Cutaneous Leishmaniasis (CL) in patients with clinical or epidemiological suspicion of CL: systematic review and meta-analysis." (#PNTD-D-19-01036) for review by PLOS Neglected Tropical Diseases. Your manuscript was fully evaluated at the editorial level and by independent peer reviewers. The reviewers appreciated the attention to an important problem, but raised some substantial concerns about the manuscript as it currently stands. These issues must be addressed before we would be willing to consider a revised version of your study. We cannot, of course, promise publication at that time.

We therefore ask you to modify the manuscript according to the review recommendations before we can consider your manuscript for acceptance. Your revisions should address the specific points made by each reviewer. 

When you are ready to resubmit, please be prepared to upload the following:

(1) A letter containing a detailed list of your responses to the review comments and a description of the changes you have made in the manuscript.

(2) Two versions of the manuscript: one with either highlights or tracked changes denoting where the text has been changed (uploaded as a "Revised Article with Changes Highlighted" file); the other a clean version (uploaded as the article file).

(3) If available, a striking still image (a new image if one is available or an existing one from within your manuscript). If your manuscript is accepted for publication, this image may be featured on our website. Images should ideally be high resolution, eye-catching, single panel images; where one is available, please use 'add file' at the time of resubmission and select 'striking image' as the file type. 

Please provide a short caption, including credits, uploaded as a separate "Other" file. If your image is from someone other than yourself, please ensure that the artist has read and agreed to the terms and conditions of the Creative Commons Attribution License at http://journals.plos.org/plosntds/s/content-license (NOTE: we cannot publish copyrighted images). 

(4) If applicable, we encourage you to add a list of accession numbers/ID numbers for genes and proteins mentioned in the text (these should be listed as a paragraph at the end of the manuscript). You can supply accession numbers for any database, so long as the database is publicly accessible and stable. Examples include LocusLink and SwissProt.

(5) To enhance the reproducibility of your results, we recommend that you deposit your laboratory protocols in protocols.io, where a protocol can be assigned its own identifier (DOI) such that it can be cited independently in the future. For instructions see http://journals.plos.org/plosntds/s/submission-guidelines#loc-methods

While revising your submission, please upload your figure files to the Preflight Analysis and Conversion Engine (PACE) digital diagnostic tool, https://pacev2.apexcovantage.com/ PACE helps ensure that figures meet PLOS requirements. To use PACE, you must first register as a user. Then, login and navigate to the UPLOAD tab, where you will find detailed instructions on how to use the tool. If you encounter any issues or have any questions when using PACE, please email us at figures@plos.org.

We hope to receive your revised manuscript by Dec 20 2019 11:59PM. If you anticipate any delay in its return, we ask that you let us know the expected resubmission date by replying to this email.

To submit a revision, go to https://www.editorialmanager.com/pntd/ and log in as an Author. You will see a menu item call Submission Needing Revision. You will find your submission record there. 

Sincerely,

Mitali Chatterjee

Guest Editor

Alain Debrabant

Deputy Editor

Reviewer #1: The objective of the study is clearly stated and the study design is well described. However, the statistical methods used are not fully discribed. For example, what statistical method was used to test differences in the accuracies.

Reviewer #2: The manuscript by Mesa et al presents a systematic review and meta-analysis regarding the test accuracy of polymerase chain reaction methods against conventional diagnostic

techniques for Cutaneous Leishmaniasis (CL) in patients with clinical or

epidemiological suspicion of CL. The authors have used both case control and consecutive studies for this purpose as the studies pertaining to this are very limited and pooled analysis of all the molecular diagnostic methods including PCR, LAMP, RPA, real time PCR were carried out.

The review protocol adopted for analysis followed by authors is standard. In conclusion from 12/13 studies from which the data could be extracted the conclusion was drawn that the molecular methods have high sensitivity and specificity in all the readout methods, and the accuracy of test is similar in smears, aspirate, skin biopsies or swabs samples. In conclusion, authors recommend that a simple smears sample run by PCR instead more invasive samples is enough to obtain a positive diagnosis of CL. 

Comments:

1. For heterogeneity analysis gold standard test were a. direct microscopy and b. microscopy and or culture. However, for generation of HSROC authors have again used gold standard A and B and here it indicates the sample type A. smear B. invasive sample.

For clarity and comparative representation fig 3 and fig 4 can be merged in to one as 3 a and 3b and it may be indicated that this analysis is about the type of sampling procedure used. As it is one of the main findings and should be emphasized from patients point of view highlighting the use of non invasive sampling procedure.

2. Line 300-302, it may be rephrased to give clear directions for the future diagnostic accuracy studies to be carried out, instead of stating that future studies will support the findings of this study. 

3. The language is not lucid at several places and interpretation become difficult, it may be improved. eg line 264; 278; 287

Reviewer #3: -Are the objectives of the study clearly articulated with a clear testable hypothesis stated? No

-Is the study design appropriate to address the stated objectives? Yes

-Is the population clearly described and appropriate for the hypothesis being tested? Yes

-Is the sample size sufficient to ensure adequate power to address the hypothesis being tested? Not applicable

-Were correct statistical analysis used to support conclusions? Yes

-Are there concerns about ethical or regulatory requirements being met? No

**Results**

-Does the analysis presented match the analysis plan?

-Are the results clearly and completely presented?

-Are the figures (Tables, Images) of sufficient quality for clarity?

Reviewer #1: Tables 1 and 2 are not available for review

line 235-236: There is no pooled estimate in Figure 2.

line 239-241: The authors wrote here and other places that "there were no significant differences in the accuracies between various methods." What statistical analysis was used to test these difference.

line 278-279: Revise the statement.

line 287: Do you mean: There was, however, "no significant" or "a non-significant" difference...

Although the results in this study are well discussed, the author almost failed to provide justifications for their findings. What are the implications for low accuracies of these tests? What are the implications?

Reviewer #2: (No Response)

Reviewer #3: -Does the analysis presented match the analysis plan? Yes, but not well presented

-Are the results clearly and completely presented? No

-Are the figures (Tables, Images) of sufficient quality for clarity? Yes.

**Conclusions**

-Are the conclusions supported by the data presented?

-Are the limitations of analysis clearly described?

-Do the authors discuss how these data can be helpful to advance our understanding of the topic under study?

-Is public health relevance addressed?

Reviewer #1: Yes

Reviewer #2: (No Response)

Reviewer #3: -Are the conclusions supported by the data presented?: No conclusions in the main manuscript!

-Are the limitations of analysis clearly described?: Somewhat

-Do the authors discuss how these data can be helpful to advance our understanding of the topic under study?: Somewhat

-Is public health relevance addressed?: Not adequately

**Editorial and Data Presentation Modifications?**

Reviewer #1: (No Response)

Reviewer #2: (No Response)

Reviewer #3: (No Response)

**Summary and General Comments**

Reviewer #1: Abstract:

1. Writes confidence interval (CI) in full only the first time used and use CI subsequently.

2. May I suggest the authors separate "Methods" and "Results"

Introduction:

3. Page 3 line 74-75: Provide reference for this statement "The evolution time of the lesion and the previous use of treatments may also interfere with the sensitivity of the direct methods."

Materials and methods

line 115: Remove the inverted question mark

line 152: I was wondering why the authors removed all studies where a 2 by 2 contingency table cannot be created?

line 211: You mean "<0.05"?

Reviewer #2: (No Response)

Reviewer #3: This paper systematically reviews and meta-analyses the sensitivity and specificity of PCR-based tests when compared to direct microscopy (+/- isolation of promastigotes in cultures) at diagnosing cutaneous leishmaniasis. It is generally sound in its methods but the writing needs more work in my opinion. Here are some comments/suggestions:

General and/or methodological comment

Abstract:

The objective and the primary conclusion don’t align. This should be corrected.

Introduction:

Perhaps the author could consider shortening it, and should clearly describe the concepts of the clinical workflow in the target condition, prior tests (like clinical suspicion), index test and comparator tests, and intended role of the index test in the clinical workflow.

No objective statement in the introduction section.

Materials and methods

Why wasn’t PRISMA-DTA not followed? This is the usual reporting standard for diagnostic systematic reviews

Was any grey literature source searched? Were conference abstracts excluded?

Search terms: I would argue including concepts of direct tests and humans restricts the search term unnecessarily ("direct exam" OR "direct test" OR "direct tests" OR "direct microscopy" OR smear)) AND Humans. How many studies were lost due to inclusion of these terms in the search term with Boolean concept AND? How many of them were relevant?

- Analysis of the heterogeneity of the papers: suggest also considering the inclusion of treated patients and study location as sources of heterogeneity between studies. 

- “Heterogeneity was assessed using Cochran’s Q test as well as Higgins’ I2 statistic”: no results eventually provided. I would suggest excluding them altogether 

Study inclusion: a number of studies have double counting of patients. Authors should be transparent about this as this means that this results in a faulty over-assumption about the variability by the statistical models.

- reconsider denoting direct test as the “gold standard”. It isn’t, indeed PCR based methods are generally thought to have better accuracy. Instead, consider using “reference standard”

Analysis:

 “Statistical analysis and meta-analysis.”: sounds odd together. Suggest just use either “Statistical analysis” or “meta-analysis”

-Suggest mentioning the model before the covariates. 

- Were the covariates entered together or individually? Since this was the formal way that the authors tested for heterogeneity, in my opinion, the former section should be called “anticipated sources of heterogeneity” and the actual assessment of heterogeneity detailed under statistical analysis.

- “heterogeneity in the studies included in meta-analysis if the P value was 0.05…”: authors do not mention any P value in the results. 

- while there is no recommended method to test for publication bias, a statement regarding this should be included. [I would also suggest considering inspection of funnel plot (diagnostic odds ratio vs sample size) and including this as a post-hoc analysis in supplementary material. I understand no inferences can be made regarding the presence or absence of publication bias though.]

Results:

“Study characteristics” usually connotes study design. Instead authors list PCR methods and source of biological material. Consider reorganizing.

“Quality assessment of study reports” logically comes before the main results (and so perhaps should the associated figures). Consider reorganizing 

Diagnostic accuracy of molecular tests and analysis of heterogeneity: consider adopting a more narrative style rather than just pointing out tables and figures.

Table 2, as presented, is very confusing. If it represents subgroup analyses, then one would intuitively think that the subgroups would be consecutively recruited vs Case-control (indeed, as stated in the analysis section), and not the combined 

categories.

Figure 1. Consider detailing the “inappropriate gold standard”s since 15 studies were excluded because of this criterion.

Why is PRISMA flow chart also included as supplementary material?

Discussion:

Generally poorly organized.

Page 10, 283-290: these are results.

Page 10, 289-290: is this analysis post-hoc? it is not mentioned in the methods

No conclusion

Other stylistic issues:

Abstract:

Background: “….PubMed, EMBASE and LILACS”: this should be part of the Methodology

Methodology/Principal findings: 

Suggest excluding "This is to be expected, because the controls in case-control studies are often healthy persons, whereas controls in consecutive studies are in fact suspected patients."

The following sentence is grammatically incorrect: “Estimates were lower on aspirates, skin biopsies and swab samples in general population 0.90 (95% confidence interval [CI], 0.80 to 0.95) and 0.87 (95% confidence interval [CI], 0.76 to 0.94) for sensitivity and specificity, respectively.” 

“More than 30 Leishmania species are recognized, of which 20 are considered infective for humans and other mammals.” This sentence should be referenced.

Other examples of grammatical errors or stylistic concerns include but not limited to:

“ significant differences between the readout method of the index test, however, subgroups were analyzed separately: “all readout methods of the index test except LAMP and real-time PCR” and “all readout methods of the index test without exception”.

“The lack of standardization is another limitation when you want to compare diagnostic accuracy studies for molecular tools.”

Page 3, 70-71

Page 4, 95

Page 7, 196-198

Page 9, 264-265

PLOS authors have the option to publish the peer review history of their article (what does this mean?). If published, this will include your full peer review and any attached files.

Reviewer #1: No

Reviewer #2: No

Reviewer #3: No

---

## [Decision Letter · Decision Letter 1]

9 Dec 2019

Dear Professor Robledo,

We are pleased to inform you that your manuscript, "Test accuracy of polymerase chain reaction methods against conventional diagnostic techniques for Cutaneous Leishmaniasis (CL) in patients with clinical or epidemiological suspicion of CL: systematic review and meta-analysis.", has been editorially accepted for publication at PLOS Neglected Tropical Diseases.

Before your manuscript can be formally accepted and sent to production you will need to complete our formatting changes, which you will receive in a follow up email. Please note: your manuscript will not be scheduled for publication until you have made the required changes.

IMPORTANT NOTES

* Copyediting and Author Proofs: To ensure prompt publication, your manuscript will NOT be subject to detailed copyediting and you will NOT receive a typeset proof for review. The corresponding author will have one final opportunity to correct any errors when sent the requests mentioned above. Please review this version of your manuscript for any errors.

* If you or your institution will be preparing press materials for this manuscript, please inform our press team in advance at plosntds@plos.org. If you need to know your paper's publication date for media purposes, you must coordinate with our press team, and your manuscript will remain under a strict press embargo until the publication date and time. PLOS NTDs may choose to issue a press release for your article. If there is anything that the journal should know, please get in touch.

*Now that your manuscript has been provisionally accepted, please log into EM and update your profile. Go to http://www.editorialmanager.com/pntd, log in, and click on the "Update My Information" link at the top of the page. Please update your user information to ensure an efficient production and billing process.

*Note to LaTeX users only - Our staff will ask you to upload a TEX file in addition to the PDF before the paper can be sent to typesetting, so please carefully review our Latex Guidelines [http://www.plosntds.org/static/latexGuidelines.action] in the meantime.

Best regards,

Mitali Chatterjee

Guest Editor

Alain Debrabant

Deputy Editor

Reviewer's Responses to Questions

**Key Review Criteria Required for Acceptance?**

**Methods**

-Are the objectives of the study clearly articulated with a clear testable hypothesis stated?

-Is the study design appropriate to address the stated objectives?

-Is the population clearly described and appropriate for the hypothesis being tested?

-Is the sample size sufficient to ensure adequate power to address the hypothesis being tested?

-Were correct statistical analysis used to support conclusions?

-Are there concerns about ethical or regulatory requirements being met?

Reviewer #2: The objectives have been stated clearly now.

Reviewer #3: (No Response)

**Results**

-Does the analysis presented match the analysis plan?

-Are the results clearly and completely presented?

-Are the figures (Tables, Images) of sufficient quality for clarity?

Reviewer #2: Yes. results are presented clearly.

Reviewer #3: (No Response)

**Conclusions**

-Are the conclusions supported by the data presented?

-Are the limitations of analysis clearly described?

-Do the authors discuss how these data can be helpful to advance our understanding of the topic under study?

-Is public health relevance addressed?

Reviewer #2: yes

Reviewer #3: (No Response)

**Editorial and Data Presentation Modifications?**

Reviewer #2: (No Response)

Reviewer #3: (No Response)

**Summary and General Comments**

Reviewer #2: (No Response)

Reviewer #3: Authors have carefully revised the manuscript. Major comments have been addressed and the paper now is more organized than the original version.

The conclusion should follow limitations section.

English still needs substantial work.eg, lines 324, 327, 350...

PLOS authors have the option to publish the peer review history of their article (what does this mean?). If published, this will include your full peer review and any attached files.

Reviewer #2: No

Reviewer #3: No

---

## [Editor Report · Acceptance letter]

13 Jan 2020

Dear Professor Robledo,

We are delighted to inform you that your manuscript, "Test accuracy of polymerase chain reaction methods against conventional diagnostic techniques for Cutaneous Leishmaniasis (CL) in patients with clinical or epidemiological suspicion of CL: systematic review and meta-analysis.," has been formally accepted for publication in PLOS Neglected Tropical Diseases.

Best regards,

Serap Aksoy

Editor-in-Chief

Shaden Kamhawi

Editor-in-Chief
